# Development of Piezo-Actuated Two-Degree-of-Freedom Fast Tool Servo System

**DOI:** 10.3390/mi10050337

**Published:** 2019-05-22

**Authors:** Yamei Liu, Yanping Zheng, Yan Gu, Jieqiong Lin, Mingming Lu, Zisu Xu, Bin Fu

**Affiliations:** School of Mechatronic Engineering, Changchun University of Technology, Changchun 130012, China; liuym_cc@126.com (Y.L.); zhengyp117@163.com (Y.Z.); lumm@ccut.edu.cn (M.L.); 15568607008@163.com (Z.X.); fb3483306540@163.com (B.F.)

**Keywords:** 2-DOF FTS system, compliant mechanism, piezoelectric actuator, matrix based compliance modeling, ultra-precision machining

## Abstract

Fast tool servo (FTS) machining technology is a promising method for freeform surfaces and machining micro-nanostructure surfaces. However, limited degrees of freedom (DOF) is an inherent drawback of existing FTS technologies. In this paper, a piezo-actuated serial structure FTS system is developed to obtain translational motions along with z and x-axis directions for ultra-precision machining. In addition, the principle of the developed 2-DOF FTS is introduced and explained. A high-rigidity four-bar (HRFB) mechanism is proposed to produce motion along the z-axis direction. Additionally, through a micro-rotation motion around flexible bearing hinges (FBHs), bi-directional motions along the x-axis direction can be produced. The kinematics of the mechanism are described using a matrix-based compliance modeling (MCM) method, and then the static analysis and dynamic analysis are performed using finite element analysis (FEA). Testing experiments were conducted to investigate the actual performance of the developed system. The results show that low coupling, proper travel, and high natural frequency are obtained. Finally, a sinusoidal wavy surface is uniformly generated by the mechanism developed to demonstrate the effectiveness of the FTS system.

## 1. Introduction

Freeform surfaces have been widely used in the aerospace, military, and biomedical engineering fields [1,2,3]. Fast tool servo (FTS)-assisted diamond cutting, because of its characteristics of high-frequency response, high positioning accuracy, and high efficiency, is widely considered to be the most promising technology for producing uniform optical freeform surfaces and microstructure surfaces [1,4,5,6]. Many scholars have made great efforts in many aspects including motion travel, working bandwidth, and trajectory-tracking accuracy to improve the performance of FTS system. Through a large amount of research, the dimensions of manufactured components range in size from millimeters [7] to micrometers [8,9] or even nanometers [4,10]. In addition, workpiece materials range from ordinary metals [11] to polymers [12], and even brittle materials (such as infrared materials and glass) [13,14], which will further demonstrate the flexibility and broad applicability of FTS technology. Chen et al. conducted precise cutting on non-planar brittle materials without knowing surface shapes beforehand by utilizing a force-controlled FTS [13].

The number of limited servo motions in existing FTS technology is inherently flawed, which limits the potential applications of FTS technology. Compared to single degrees of freedom (DOF) FTS, multi-DOF FTS is more flexible, enabling multi-directional active cutting and displacement compensation [15,16]. It ensures synchronization and coordination of the motion associated with the spindle to improve the capability of generating complicated surfaces and microstructures [17]. Therefore, the development of multi-DOF FTS is an inevitable trend.

There are two operating modes of vibration-assisted cutting: resonant mode vibration cutting and non-resonant mode vibration cutting. Resonant vibration cutting is a cutting method that realizes surface machining of a workpiece by applying high-frequency and small-amplitude vibration in the direction of tool movement. Intermittent contact between the tool and the workpiece during resonant vibration cutting improves dynamic cutting stability and reduces cutting forces. Guo and Ehmann et al. developed a three-motion generator with ultrasonic vibration to quickly generate a textured surface. Although the cutting mode based on resonant vibration has higher efficiency [18,19], its operating frequency is relatively simple and can only work at certain discrete frequencies. Therefore, it is less adaptable to the machining of certain complex surfaces. In contrast, non-resonant mode vibration cutting is typically accomplished using a piezo-driven flexure mechanism that can operate at a range of continuous frequencies. The frequency and amplitude of the output trajectory can be adjusted flexibly during non-resonant vibration cutting operations, so this machining method provides greater flexibility in the machining of certain complex and difficult-to-machine surfaces [20,21]. However, for the current research, the non-resonant vibration mode cutting has a lower operating frequency, resulting in lower machining efficiency and poor surface quality. Therefore, most scholars have made a lot of efforts to develop high-frequency non-resonant cutting devices. Wada et al. developed a three-axis controlled fast tool servo system for machining freeform surfaces and microstructures, reducing the cutting time and increasing machining efficiency [22]. However, the strong coupling of the two ends of the mechanism will result in undesirable parasitic motion. A new 3-PUU (3 represents three branched chains, P represents prismatic pair, U represents hook hinge) fast tool servo system with a large working stroke and high precision was proposed by Tang et al. for machining complex microstructure. However, it had a low natural frequency [15].

Compared to three-axis motion, the 2-DOF device has better stability and can provide higher machining accuracy. Zhu et al. designed a parallel structure 2-DOF FTS by introducing a z-shaped flexure hinge to generate the servo motions with decoupled motions along with the x and z-axis directions. However, the device was tested for lower bandwidth in the experiment due to incomplete contact and inertial movement [23].

At present, multi-DOF non-resonant vibration cutting technology mainly adopts a piezoelectric-driven parallel structure-compliant mechanism to obtain higher operating frequency and stiffness [5,24]. However, most of the parallel mechanisms have a strong coupling effect, and the motion inaccuracy caused by the coupling may cause the shape accuracy of the generated surface to deteriorate. A hybrid macro-and micro-range FTS device, driven by a voice coil motor (VCM) and a piezoelectric actuator, respectively, was proposed by Liu et al. [7]. The developed device adopted an approximate serial structure to achieve appropriate travel and low resolution, however, a lower bandwidth was obtained due to the inertia of the motion. 

In this paper, a FTS system with serial structure is designed, which has the characteristics of simple control and independent motion. In addition, it has good flexibility, and it can extend the processing range of complex surfaces due to its active cutting motion along both the x and z-axis directions. The rest of this article is organized as follows: The structure design is introduced in Section 2. In Section 3, the matrix-based compliance modeling method is used to analyze the kinematics of the developed device, and finite element analysis (FEA) is used for static analysis and dynamic analysis. Then, the prototype is manufactured and tested for studying the practical performance of the mechanism in Section 4. In Section 5, the effectiveness of the developed device is verified by cutting experiments.

## 2. Mechanical Design of Two Degrees of Freedom Fast Tool Servo (2-DOF FTS)

The serial structure is adopted to achieve motions in the x and z-axis directions, due to its simple control and independent movement. The mechanical structure of the developed mechanism is illustrated in Figure 1.

For motion along the z-axis direction, the high rigidity four bar (HRFB) mechanism with four right circular flexure hinges (RCFHs) and four leaf spring flexible hinges (LSFHs) [23,25,26] is employed to isolate and guide the movement of the PFA as shown in Figure 2a. The motion along the x-axis direction is achieved by a micro-ration motion generated by a flexible bearing hinges (FBHs) [27] with three RCFHs connected in parallel as illustrated in Figure 2b, which can be approximated as a linear motion due to the extremely small motion stroke. The x-axis direction piezoelectric is driven and guided through a guiding mechanism consisting of RCFHs as illustrated in Figure 2c. The inclined LSFHs are connected in parallel to provide a restoring force for the micro-ration motion generated by the FBHs in the x-axis direction as shown in Figure 2d. The symmetrical design is employed to increase the motion accuracy and decrease the coupling effect. The corresponding motion principles of each part of the device will be elaborated in Section 3.

The mechanical structure and motion principles of HRFB are illustrated in Figure 3. Normally, the bending deformation only occurs on the flexible hinges. As shown in Figure 3, when the force F_Z_ push the input point $O_{p}$, two combination linkages M12 and M34 will bend to generate the displacements $\Delta_{z_{1}}$ and $\Delta_{z_{2}}$. The displacement $\Delta_{z}$ produced by the combination of the displacements $\Delta_{z_{1}}$ and $\Delta_{z_{2}}$ will cause the mechanism to move in the z-axis direction.

Figure 4 shows a simplified diagram of the FBHs motion principle. It can be seen that the FBHs rotate around the center point, providing the mechanism with lateral displacement along the x-axis direction. 

The configuration surrounded by three sides is adopted in the device to ensure that it has a higher rigidity in the cutting direction. It can reduce the deformation of the FTS device caused by the cutting force to improve the machining accuracy. In general, the mechanism is designed to be symmetrical about the z-axis direction in order to reduce the effects of lateral motion and coupling during the cutting process [28]. When the PEA (piezoelectric actuator) applies signals to the guiding mechanism, the HRFB will move along the z-axis direction while the right guiding mechanism and FBHs are driven by the right PEA to produce a micro-rotational motion along the x-axis direction.

## 3. Compliance Modeling of 2-DOF FTS

### 3.1. Compliance Modeling Based on Matrix Method

The elastic mechanics model of the basic flexible unit can be obtained by the theory of screw variables. When a load is applied to the end of the flexible unit, the end of the flexible unit is deformed or slightly moved. According to the screw theory [29], the deformation of the end of the flexible unit and the load applied to the hinge can be represented by the motion spin amount $U={\left(\begin{array}{cccccc} \delta_{x} & \delta_{y} & \delta_{z} & \theta_{x} & \theta_{y} & \theta_{z} \end{array}\right)}^{T}$ and the force spin amount $W={\left(\begin{array}{cccccc} F_{x} & F_{y} & F_{z} & M_{x} & M_{y} & M_{z} \end{array}\right)}^{T}$, respectively.

Equation (1) shows the relationship between the externally applied load and end deformation, where the flexibility factor $C_{ij}$ can be determined by the flexible hinge size parameters [30].(1)$$\left[\begin{array}{l} \delta_{x}\\ \delta_{y}\\ \delta_{z}\\ \theta_{x}\\ \theta_{y}\\ \theta_{z} \end{array}\right]=\left[\begin{array}{cccccc} c_{11} & 0 & 0 & 0 & c_{15} & 0\\ 0 & c_{22} & 0 & c_{24} & 0 & 0\\ 0 & 0 & c_{33} & 0 & 0 & 0\\ 0 & c_{42} & 0 & c_{44} & 0 & 0\\ c_{51} & 0 & 0 & 0 & c_{55} & 0\\ 0 & 0 & 0 & 0 & 0 & c_{66} \end{array}\right]\left[\begin{array}{c} F_{x}\\ F_{y}\\ F_{z}\\ M_{x}\\ M_{y}\\ M_{z} \end{array}\right],$$

The basic matrix of the RCFHs is obtained by the traditional computational model developed by P&W (Paros and Weisbord) [31]. The dimensional parameters of the LSFHs and RCFHs used in this paper are shown in Figure 5.

The complete compliance matrix needs to perform a coordinate transformation from each local coordinate system to a unified global coordinate system. The transformation relationship of the compliance matrix from the local coordinate system $O_{i}$ to the global coordinate system $O_{j}$ can be expressed as:(2)$$C_{i}^{j}=T_{i}^{j}C_{i}{\left(T_{i}^{j}\right)}^{T},$$

The transformation matrix is shown in Equation (3)(3)$$T_{i}^{j}=\left[\begin{array}{cc} R_{i}^{j} & S(r_{i}^{j})R_{i}^{j}\\ 0 & R_{i}^{j} \end{array}\right],$$
where $R_{i}^{j}$ is the rotation transformation matrix of coordinate $O_{i}$ with respect to $O_{j}$, $r_{i}^{j}$ is the position vector of a point $O_{i}$ expressed in the global coordinate system $O_{j}$, S(r) represents the skew-asymmetric matrix of the translation vector $r={\left[\begin{array}{ccc} r_{x} & r_{y} & r_{z} \end{array}\right]}^{T}$ from the local coordinate system $O_{i}$ to the global coordinate system $O_{j}$, and 0 is the third-order zero square matrices. The matrix S(r) can be obtained by:(4)$$S\left(r\right)=\left[\begin{array}{ccc} 0 & -r_{z} & r_{y}\\ r_{z} & 0 & -r_{x}\\ -r_{y} & r_{x} & 0 \end{array}\right],$$

The matrix $R_{i}^{j}$ can be written as:(5)$$R_{i}^{j}=\left[\begin{array}{ccc} \cos\theta & 0 & -\sin\theta \\ 0 & 1 & 0\\ \sin\theta & 0 & \cos\theta \end{array}\right],$$
where θ is the angle of rotation around the y-axis.

### 3.2. Output Compliance Modelling

As shown in Figure 2, the FTS mechanism can be viewed as consisting of the following six parts and labeled as the module I, II, III, IV, V, VI, respectively. The output compliance $C_{}^{O_{\mathrm{out}}}$ of each module is considered to be the compliance at the output point $O_{\mathrm{out}}$. 

#### 3.2.1. Output Compliance of the Module I

As for the module I, it mainly consists of an RCFH 5 and two linkages A1 and A2, as shown in Figure 6. Each of the linkages can be considered to be connected in series by two RCFHs.

According to the MCM method, the compliance of linkages A1 and A2 at the point $O_{\mathrm{out}}$ can be obtained from: (6)$$C_{A1}^{O_{\mathrm{out}}}=C_{1}^{O_{\mathrm{out}}}+C_{2}^{O_{\mathrm{out}}}=T_{1}^{O_{\mathrm{out}}}C_{1}^{R}{\left(T_{1}^{O_{\mathrm{out}}}\right)}^{T}+T_{2}^{O_{\mathrm{out}}}C_{2}^{R}{\left(T_{2}^{O_{\mathrm{out}}}\right)}^{T},$$
(7)$$C_{A2}^{O_{\mathrm{out}}}=C_{3}^{O_{\mathrm{out}}}+C_{4}^{O_{\mathrm{out}}}=T_{3}^{O_{\mathrm{out}}}C_{3}^{R}{\left(T_{3}^{O_{\mathrm{out}}}\right)}^{T}+T_{4}^{O_{\mathrm{out}}}C_{4}^{R}{\left(T_{4}^{O_{\mathrm{out}}}\right)}^{T},$$
where $T_{i}^{O_{\mathrm{out}}}$ (i = 1, 2, 3, 4) denote the compliance transformation matrix (CTM) from the local coordinate of RCFHs to the coordinate $O_{\mathrm{out}}-xz$, and $C_{i}^{R}$ denotes the initial compliance matrix of RCFHs in its local coordinates.

Because two linkages A1 and A2 are connected in parallel, and then serial with a separate RCFH 5, the compliance of module I in the coordinate $O_{\mathrm{out}}-xz$ can be derived as: (8)$$C_{A}^{O_{\mathrm{out}}}={\left[{\left(C_{A1}^{O_{\mathrm{out}}}\right)}^{-1}+{\left(C_{A2}^{O_{\mathrm{out}}}\right)}^{-1}\right]}^{-1}+T_{5}^{O_{\mathrm{out}}}C_{5}^{R}{\left(T_{5}^{O_{\mathrm{out}}}\right)}^{T},$$

Based on the symmetry of module I and IV with respect to the z-axis, as shown in Figure 2. The compliance $C_{B}^{O_{\mathrm{out}}}$ of module IV can be obtained through rotating the compliance $C_{A}^{O_{\mathrm{out}}}$
(9)$$C_{B}^{O_{\mathrm{out}}}=T_{rz}^{O_{\mathrm{out}}}\left(\pi \right)C_{A}^{O_{\mathrm{out}}}{\left[T_{rz}^{O_{\mathrm{out}}}\left(\pi \right)\right]}^{T},$$
where $T_{rz}\left(\pi \right)$ represents the rotation CTM around z-axis with an angle of π.

#### 3.2.2. Output Compliance of the Module II

With module II, it mainly consists of two parallel LSFHs, and its compliance can be expressed as(10)$$C_{C}^{O_{\mathrm{out}}}={\left[{\left(C_{6}^{O_{\mathrm{out}}}\right)}^{-1}+{\left(C_{7}^{O_{\mathrm{out}}}\right)}^{-1}\right]}^{-1}={\left\{{\left[T_{6}^{O_{\mathrm{out}}}C_{6}^{L}{\left(T_{6}^{O_{\mathrm{out}}}\right)}^{T}\right]}^{-1}+{\left[T_{7}^{O_{\mathrm{out}}}C_{7}^{L}{\left(T_{7}^{O_{\mathrm{out}}}\right)}^{T}\right]}^{-1}\right\}}^{-1},$$
where $T_{i}^{O_{\mathrm{out}}}$ (i = 6, 7) represent the compliance transformation matrix (CTM) from the local coordinate of the LSFHs to the coordinate $O_{\mathrm{out}}-xz$, and $C_{i}^{L}$ denotes the initial compliance of LSFHs in its local coordinates. 

Similar to that obtained in Equation (9), the compliance of module III can be obtained by rotating compliance $C_{C}^{O_{\mathrm{out}}}$ at an angle of π around the z-axis, as shown in Figure 2.(11)$$C_{D}^{O_{\mathrm{out}}}=T_{rz}^{O_{\mathrm{out}}}\left(\pi \right)C_{C}^{O_{\mathrm{out}}}{\left[T_{rz}^{O_{\mathrm{out}}}\left(\pi \right)\right]}^{T},$$

#### 3.2.3. Output Compliance of the Module V

The FBHs can be seen as a parallel connection of three RCFHs, as shown in Figure 7. The principle of the structure is changing the original biased displacement flexible bearing into a 90-degree distribution to achieve the rotation effect. It can meet the requirement of high frequency and rigidity of the device, and also prevents the shaft drift. The compliance of the FBHs can be written as:(12)$$\begin{array}{l} C_{E}^{O_{\mathrm{out}}}={\left[{\left(C_{8}^{O_{\mathrm{out}}}\right)}^{-1}+{\left(C_{9}^{O_{\mathrm{out}}}\right)}^{-1}+{\left(C_{10}^{O_{\mathrm{out}}}\right)}^{-1}\right]}^{-1}\\ ={\left\{{\left[T_{8}^{O_{\mathrm{out}}}C_{8}^{R}{\left(T_{8}^{O_{\mathrm{out}}}\right)}^{T}\right]}^{-1}+{\left[T_{9}^{O_{\mathrm{out}}}C_{9}^{R}{\left(T_{9}^{O_{\mathrm{out}}}\right)}^{T}\right]}^{-1}+{\left[T_{10}^{O_{\mathrm{out}}}C_{10}^{R}{\left(T_{10}^{O_{\mathrm{out}}}\right)}^{T}\right]}^{-1}\right\}}^{-1} \end{array}$$
where $T_{i}^{O_{\mathrm{out}}}$ (i = 8, 9, 10) denotes the compliance transformation matrix (CTM) from the local coordinate of the RCFHs to the coordinate $O_{\mathrm{out}}-xz$.

#### 3.2.4. Output Compliance of Module VI

The HRFB is a high stiffness group including two combination linkages M12 and M34 consisting of four RCFHs and four LSFHs, respectively. As shown in Figure 8, the small linkages M1 and M2 are, respectively, composed of two RCFHs connected in parallel. A combination linkage M12 is connected in serial by the small linkages M1 and M2. Hence, compliance $C_{M12}^{O_{\mathrm{out}}}$ with respect to the coordinate $O_{\mathrm{out}}-xz$ system can be expressed as: (13)$$\begin{array}{l} C_{M12}^{O_{\mathrm{out}}}=C_{M1}^{O_{\mathrm{out}}}+C_{M2}^{O_{\mathrm{out}}}={\left[{\left(C_{11}^{O_{\mathrm{out}}}\right)}^{-1}+{\left(C_{12}^{O_{\mathrm{out}}}\right)}^{-1}\right]}^{-1}+{\left[{\left(C_{13}^{O_{\mathrm{out}}}\right)}^{-1}+{\left(C_{14}^{O_{\mathrm{out}}}\right)}^{-1}\right]}^{-1}\\ ={\left\{{\left[T_{11}^{O_{\mathrm{out}}}C_{11}^{R}{\left(T_{11}^{O_{\mathrm{out}}}\right)}^{T}\right]}^{-1}+{\left[T_{12}^{O_{\mathrm{out}}}C_{12}^{R}{\left(T_{12}^{O_{\mathrm{out}}}\right)}^{T}\right]}^{-1}\right\}}^{-1}+{\left\{{\left[T_{13}^{O_{\mathrm{out}}}C_{13}^{R}{\left(T_{13}^{O_{\mathrm{out}}}\right)}^{T}\right]}^{-1}+{\left[T_{14}^{O_{\mathrm{out}}}C_{14}^{R}{\left(T_{14}^{O_{\mathrm{out}}}\right)}^{T}\right]}^{-1}\right\}}^{-1} \end{array}$$

Similar to that obtained in Equation (13), the compliance $C_{M34}^{O_{\mathrm{out}}}$ of another combination linkage M34 can be obtained in the $O_{\mathrm{out}}-xz$ system as: (14)$$\begin{array}{l} C_{M34}^{O_{\mathrm{out}}}=C_{M3}^{O_{\mathrm{out}}}+C_{M4}^{O_{\mathrm{out}}}={\left[{\left(C_{15}^{O_{\mathrm{out}}}\right)}^{-1}+{\left(C_{16}^{O_{\mathrm{out}}}\right)}^{-1}\right]}^{-1}+{\left[{\left(C_{17}^{O_{\mathrm{out}}}\right)}^{-1}+{\left(C_{18}^{O_{\mathrm{out}}}\right)}^{-1}\right]}^{-1}\\ ={\left\{{\left[T_{15}^{O_{\mathrm{out}}}C_{15}^{L}{\left(T_{15}^{O_{\mathrm{out}}}\right)}^{T}\right]}^{-1}+{\left[T_{16}^{O_{\mathrm{out}}}C_{16}^{L}{\left(T_{16}^{O_{\mathrm{out}}}\right)}^{T}\right]}^{-1}\right\}}^{-1}+{\left\{{\left[T_{17}^{O_{\mathrm{out}}}C_{17}^{L}{\left(T_{17}^{O_{\mathrm{out}}}\right)}^{T}\right]}^{-1}+{\left[T_{18}^{O_{\mathrm{out}}}C_{18}^{L}{\left(T_{18}^{O_{\mathrm{out}}}\right)}^{T}\right]}^{-1}\right\}}^{-1} \end{array}$$

The two combination linkages M12 and M34 are connected in parallel with respect to the point $O_{\mathrm{out}}$ to form a high stiffness group, and its compliance can be expressed as:(15)$$C_{M}^{O_{\mathrm{out}}}={\left[{\left(C_{M12}^{O_{\mathrm{out}}}\right)}^{-1}+{\left(C_{M34}^{O_{\mathrm{out}}}\right)}^{-1}\right]}^{-1},$$

According to the above discussion, the complete output compliance of the mechanism can be expressed as a parallel connection of output compliance of module I, II, III, IV, V, and then in serial with module VI.(16)$$C_{\mathrm{out}}^{O}={\left[{\left(C_{A}^{O_{\mathrm{out}}}\right)}^{-1}+{\left(C_{B}^{O_{\mathrm{out}}}\right)}^{-1}+{\left(C_{C}^{O_{\mathrm{out}}}\right)}^{-1}+{\left(C_{D}^{O_{\mathrm{out}}}\right)}^{-1}+{\left(C_{E}^{O_{\mathrm{out}}}\right)}^{-1}\right]}^{-1}+C_{M}^{O_{\mathrm{out}}},$$

### 3.3. Input Compliance Modelling

For the z-axis direction input stiffness is only related to the HRFB, so only the HRFB is considered. Without considering other parts, the compliance model of the HRFB is established to obtain input compliance in the z-axis direction as shown in Figure 9a.

The model can be simplified as shown in Figure 9b, Where $C_{L}^{O_{P}}$ and $C_{R}^{O_{P}}$ denote the compliance of left and right parts of the HRFB mechanism with respect to coordinate $O_{P}-xz$, and it can be derived as:(17)$$\begin{array}{l} C_{L}^{O_{P}}={\left[{\left(C_{17}^{O_{P}}+C_{13}^{O_{P}}\right)}^{-1}+{\left(C_{18}^{O_{P}}+C_{14}^{O_{P}}\right)}^{-1}\right]}^{-1}\\ ={\left\{{\left[T_{17}^{O_{P}}C_{17}^{L}{\left(T_{17}^{O_{P}}\right)}^{-1}+T_{13}^{O_{P}}C_{13}^{R}{\left(T_{13}^{O_{P}}\right)}^{-1}\right]}^{-1}+{\left[T_{18}^{O_{P}}C_{18}^{L}{\left(T_{18}^{O_{P}}\right)}^{-1}+T_{14}^{O_{P}}C_{14}^{R}{\left(T_{14}^{O_{P}}\right)}^{-1}\right]}^{-1}\right\}}^{-1} \end{array}$$
(18)$$\begin{array}{l} C_{R}^{O_{P}}={\left[{\left(C_{15}^{O_{P}}+C_{11}^{O_{P}}\right)}^{-1}+{\left(C_{16}^{O_{P}}+C_{12}^{O_{P}}\right)}^{-1}\right]}^{-1}\\ ={\left\{{\left[T_{15}^{O_{P}}C_{15}^{L}{\left(T_{15}^{O_{P}}\right)}^{-1}+T_{11}^{O_{P}}C_{11}^{R}{\left(T_{11}^{O_{P}}\right)}^{-1}\right]}^{-1}+{\left[T_{16}^{O_{P}}C_{16}^{L}{\left(T_{16}^{O_{P}}\right)}^{-1}+T_{12}^{O_{P}}C_{12}^{R}{\left(T_{12}^{O_{P}}\right)}^{-1}\right]}^{-1}\right\}}^{-1} \end{array}$$

By combining Equations (17) and (18), the input compliance in the z-axis direction can be obtained from: (19)$$C_{\mathrm{in}}^{O_{P}}={\left[{\left(C_{L}^{O_{P}}\right)}^{-1}+{\left(C_{R}^{O_{P}}\right)}^{-1}\right]}^{-1},$$

Since the x-axis direction input compliance is independent of the HRFB mechanism, it is not considered in the calculation. Therefore, the following compliance model is established to obtain input compliance in the x- axis direction, as shown in Figure 10.

Where $C_{L}^{O_{t}}$, $C_{N}^{O_{t}}$, $C_{O}^{O_{t}}$, $C_{P}^{O_{t}}$, $C_{Q}^{O_{t}}$ are the compliance of modules I, II, III, IV, V, respectively, with respect to the coordinate system $O_{t}-xz$.

Where,(20)$$C_{L}^{O_{t}}={\left[{\left(C_{1}^{O_{t}}+C_{2}^{O_{t}}\right)}^{-1}+{\left(C_{3}^{O_{t}}+C_{4}^{O_{t}}\right)}^{-1}\right]}^{-1}+C_{5}^{O_{t}},$$
(21)$$C_{N}^{O_{t}}={\left[{\left(C_{6}^{O_{t}}+C_{7}^{O_{t}}\right)}^{-1}\right]}^{-1},$$
(22)$$C_{O}^{O_{t}}={\left[{\left(C_{8}^{O_{t}}\right)}^{-1}+{\left(C_{9}^{O_{t}}\right)}^{-1}+{\left(C_{10}^{O_{t}}\right)}^{-1}\right]}^{-1}$$
(23)$$C_{P}^{O_{t}}={\left[{\left(C_{19}^{O_{t}}+C_{20}^{O_{t}}\right)}^{-1}+{\left(C_{21}^{O_{t}}+C_{22}^{O_{t}}\right)}^{-1}\right]}^{-1}+C_{23}^{O_{t}},$$
(24)$$C_{Q}^{O_{t}}={\left[{\left(C_{24}^{O_{t}}+C_{25}^{O_{t}}\right)}^{-1}\right]}^{-1},$$
where $C_{i}^{O_{t}}$
(i=1∼10,19∼25) represent the compliance matrix of RCFHs and LSFHs in the coordinate system $O_{t}-xz$. Then, the input compliance in the x-axis direction can be derived as follows: (25)$$C_{\mathrm{in}}^{O_{t}}={\left[{\left(C_{N}^{O_{t}}\right)}^{-1}+{\left(C_{O}^{O_{t}}\right)}^{-1}+{\left(C_{P}^{O_{t}}\right)}^{-1}+{\left(C_{Q}^{O_{t}}\right)}^{-1}\right]}^{-1}+C_{M}^{O_{t}},$$

### 3.4. Finite Element Analysis of Mechanism

The selected material for the mechanism is a new material 7075 Al with elasticity modulus = 71.7 GPa, yield strength = 503 MPa, Poisson’s ratio = 0.33, and density = 2810 kg/m^3^. The nominal stroke and stiffness of the PEA are *d*_0_ = 32 µm and *k*_pea_ = 35 N·µm^−1^, respectively. To verify the analytical model, FEA through the finite element software ABAQUS is conducted, and the 3D tetrahedron element is used to mesh the model.

#### 3.4.1. Static Analysis

The overall dimensional parameters of the FTS mechanism in three directions and the material 7075 Al parameters used to simulate the stresses in the static analysis are listed in Table 1. During the analysis, the size parameters of HRFB (part VI) are set as shown in Table 2, and the parameters of other parts are set as shown in Table 3. 

In order to prevent failure in the application of the flexible mechanism, the maximum stress of the thinnest part of the flexible mechanism must be lower than the yield stress of the material (503 MPa). The input force *F* = [1000 N, 1000 N]*^T^* was applied to both the x- and z-axis input end to test the maximum stress of the developed mechanism. When subjected to the maximum input displacements in both directions, the equivalent stress distribution obtained by FEA indicates that the maximum stress appears at the position shown in Figure 11, which is approximately 214.8 MPa. It is much lower than the yield stress 503 MPa of the material, indicating that reproducible elastic deformation can be obtained during processing.

As shown in Table 4, the input stiffness in the z-axis direction is 31.33 N/μm, which is a 10.16% error with the theoretical value obtained by the compliance matrix method. The input stiffness in the x-axis direction obtained by finite element analysis is 18.12 N/μm, which is a 13.09 % error compared with the theoretical value obtained by the compliant matrix method. In the z-axis direction, the output stiffness is 15.32 N/μm, with an error of 25.06% from the theoretical value. Smaller deviations indicate the accuracy and reliability of the structural analysis model developed.

#### 3.4.2. Dynamic Analysis

Modal analysis of the mechanism is performed by FEA to obtain dynamic performance. The first four modes are shown in Figure 12. It can be seen that the first mode rotates around the FBHs, and the second mode reciprocating vibrates along the z-axis. As shown in Figure 12a,b, the corresponding natural frequencies for the first two modes with 1463.7 and 1501.6 Hz, respectively.

Furthermore, vibration and torsional motions in the y-axis direction occur in the third and fourth modes, and the respective natural frequencies are 1808.5 Hz and 1910.3 Hz, respectively, as shown in Figure 12c,d. It can be seen that these two frequencies are higher than the first two frequencies. Therefore, the first two modes are the most dominant modes. However, the dynamic performance of the FTS during actual cutting also requires consideration of the effects of piezoelectricity in both directions and the components used in the assembly on the overall stiffness of the mechanism. Therefore, we performed a dynamic analysis of the FTS assembly (considering piezoelectric actuators and assembly components) to define the expected excitation frequency during the actual cutting. Figure 13 shows the first four natural frequencies and modes of the FTS assembly in the dynamic analysis. 

As shown in Figure 13a,b, the corresponding natural frequencies of the first two modes are 1214.1 Hz and 1930.4 Hz, respectively. Therefore, in dynamic analysis, the expected excitation frequency during actual cutting is approximately 1200 Hz. In addition, the vibration mode of the FTS assembly including the piezoelectric actuator can also be seen in the simulation. The first vibration mode of the FTS assembly is basically consistent with the first vibration mode of the flexible mechanism, and the third vibration mode of the FTS assembly is basically consistent with the second vibration mode of the flexible mechanism. In general, the vibration mode does not change much.

## 4. Performance Test of the 2-DOF FTS

### 4.1. Experimental Setup 

An off-line experimental test system was constructed to conduct a comprehensive study of the proposed device. The multi-axis controller PMAC (Delta Tau Inc., Chatsworth, CA, USA) was used to generate and collect control signals during the test. The power amplifier (E-500, PI Inc., Auburn, MA, USA) was used to amplify the command signals driving the PEAs (40VS12, Harbin Core Tomorrow Science &Technology Co., Ltd., Harbin, China). Two PZT actuators of size *φ* 12 × 51.5 mm are embedded in the structure to drive the motion of the device. The nominal stroke and stiffness of the PEA are *d*_0_ = 32 µm and *k*_pea_ = 35 N·µm^−1^, respectively. The displacement of the prototype was measured by a capacitive sensor with four channels. The experimental setup is shown in Figure 14.

### 4.2. Experimental Results 

#### 4.2.1. Stroke and Decoupling Tests

To investigate the practical stroke along with two directions, it is necessary to study the maximum displacements of both z-axis direction and x-axis direction. In addition, the corresponding coupling motion along the x-axis direction and z-axis direction are also studied, respectively. 

As shown in Figure 15a and Figure 16a, the strokes obtained are about 17 µm and 22.5 µm along the z and x-axis direction, respectively. As shown in Figure 15b, the induced coupling along the x-axis direction is approximately 0.25 μm, which is 1.47% of the stroke along the z-axis direction. As shown in Figure 16b, the induced coupling along the z-axis direction is observed to be 0.5 µm, which is approximately 2.22% of the stroke along the x-axis direction. The relatively small coupling ratio may be induced by a machining error or misalignment of the actuation shaft. However, as shown in Figure 15a and Figure 16a, the command and response line are not exactly the same. The possible reason is the friction between the flexure-based FTS device and the fixed base plate, which is caused by manufacturing accuracy and assembly error. If the contact surface is sufficiently smooth, the actual response displacement can be closer to the command displacement result. In addition, the maximum input command cannot be tracked due to the force and displacement of the piezoelectric input being limited. 

#### 4.2.2. Dynamic Performance Tests

The swept excitation method was used to analyze the dynamic performance of the developed device. The measured results along the z-axis direction and the x-axis direction are displayed in Figure 17, the natural frequencies are approximately 1201 and 1220 Hz, respectively. The low bandwidth in the experiment may be due to the increase in the moving inertia caused by incomplete contact compared to the FEA results. The natural frequency of the device obtained by the sweep excitation method is reduced by about 200 Hz compared to the natural frequency obtained in the dynamic analysis of the flexible mechanism due to the influence of the piezoelectric and assembly components. However, the results obtained by the sweep test are almost identical to those obtained by the dynamic analysis of the FTS assembly (taking into account the piezoelectric and assembled components). Moreover, the discrepancy mainly caused by the added mass and imperfect contacts between PEAs and input ends. Minimizing manufacturing errors can also make sweep test results closer to dynamic analysis results.

#### 4.2.3. Resolution Tests

Resolution is one of the key criteria for achieving high-precision machining of the cutting tool. In this paper, a stair-step command signal was generated by utilizing a digital computer, through a D/A converter board, which was supplied to the PEAs to drive the FTS. As shown in Figure 18, the resolution of the z-axis direction and the x-axis direction are measured by high-precision capacitive sensors, which are approximately 45 nm and 50 nm, respectively.

#### 4.2.4. Step Responses Tests

To investigate the tool-positioning performance, the step responses along the two directions are examined, resulting in the data shown in Figure 19. A typical proportional-integral-derivative (PID) controller was implemented to position the FTS device. As shown in Figure 19, the rise times of the z-axis and x-axis motions are approximately 37 ms and 50 ms, respectively. Moreover, no steady errors and no overshoots are observed during the positioning process.

#### 4.2.5. Hysteresis Analysis

Due to the inherent defect hysteresis of PEA, it has a great influence on the motion trajectory. Therefore, this paper analyzes the hysteresis of the strokes of the developed FTS in detail. In order to study the hysteresis of the developed FTS device, a triangular voltage with 10 V amplitude and 3 Hz frequency generated by the signal generator is amplified by a charge amplifier and applied to the two PEAs in the z- and x-axis directions, respectively. Figure 20 shows the input signal and the corresponding input voltage-displacement response curve. As shown in Figure 20, the lower curve shows the expansion of the PEA, and the upper curve shows the retraction of the PEA. It can be seen that the maximum positioning differences between the expansion and retraction of the two axes of motion are 1.67 μm and 1.75 μm, respectively, which are about 9.2% and 9.7% of the displacement at 100 V input voltage, respectively. Therefore, in future work, we need to adopt appropriate control strategies to reduce the hysteresis to improve the motion accuracy of the trajectory.

## 5. Processing Performance Verification of the Developed 2-DOF FTS

### 5.1. Experimental Setup

The newly developed 2-DOF FTS system was integrated into the ultra-precision machine tool (Nanoform 250, AMETEK Precitech, Inc., Keene, NH, USA) for cutting experiments to verify the effectiveness of the developed device. As shown in Figure 21, the FTS device was fixed on the dynamometer of the z-axis direction guide. The workpiece was mounted on the spindle and rotates with the spindle. For fast servo motion along the z-axis, it was used to provide micro-feeding based on the feed motion of the machine’s z-axis guide to produce the same surface as a conventional FTS. The fast servo motion along the x-axis was used to provide micro-feeding based on the feed motion of the machine’s x-axis guide to increase FTS machining flexibility. Compared to most single-DOF FTS and systems that use only machine controllers, the FTS system developed in this paper can provide additional vibration along the x-axis of the machine to better match the machine’s motion. In addition, compared to the previously developed 2-DOF FTS, the experimental setup of this paper can provide high-bandwidth servo motion along the x and z-axis of the machine tool to create a surface.

### 5.2. Results and Discussions

A brass rod having a diameter of 12.7 mm was used as the workpiece. A polycrystalline diamond (PCD) tool with a nose radius of 2 mm, a rake angle of 0° and a relief angle of 10° was employed as a cutting tool. A sinusoidal signal with an amplitude of 15 μm and a vibration frequency of 200 Hz was applied to the x- and z-axis directions of the PEA, respectively. The phase difference between the x- and z-axis directions is 90°. Figure 22a shows the simulated sinusoidal wavy surface, and the practical machined surfaces by the developed FTS is shown in Figure 22b. 

The surface topography was measured by an optical surface profiler (Zygo Newview, Middlefield, CT, USA). Figure 23 shows the sinusoidal surface topography. The roughness of the machined surfaces is Sa = 24.426 μm. For ultra-precision machining, the large roughness may be due to the large vibration of the tool during cutting. Fortunately, the feasibility and effectiveness of the proposed FTS device are verified by experiments. In future work, we will make more efforts to improve the machining results.

## 6. Conclusions

In this paper, a serial piezoelectric-driven FTS system has been developed that enables motion in both z- and x-axis directions. The movement principle of the developed 2-DOF FTS is introduced. The prototype is then machined, and the actual performance of the device such as practical stroke, decoupling behavior, and natural frequency are tested. The main conclusions can be drawn as follows:(1)The input and output compliances of the developed device are analyzed and calculated using the matrix-based modeling method. In addition, the theoretical results are verified by FEA analysis. The results show that there is good agreement between theoretical analysis and FEA analysis, which proves the effectiveness of the design process. (2)Experimental tests show that the stroke along the z-axis direction and the x-axis direction can reach 17 μm and 22.5 μm, respectively, and the resolution can reach 45 nm and 50 nm, respectively. The coupling motions in both directions are measured to be approximately 1.47% and 2.22%, respectively. It is also capable to achieve high first natural frequencies. Using the swept excitation test, the natural frequencies of the FTS along the z-axis direction and the x-axis direction can reach 1201 Hz and 1220 Hz, respectively. (3)Finally, the sinusoidal wavy surface processing experiment validates the effectiveness of the developed device for ultra-precision machining.

## Figures and Tables

**Figure 1 micromachines-10-00337-f001:**
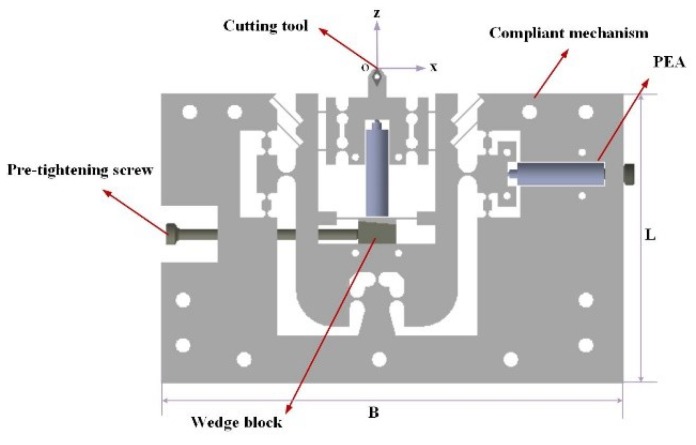
The mechanical structure of the 2-DOF (degrees of freedom) fast tool servo (FTS).

**Figure 2 micromachines-10-00337-f002:**
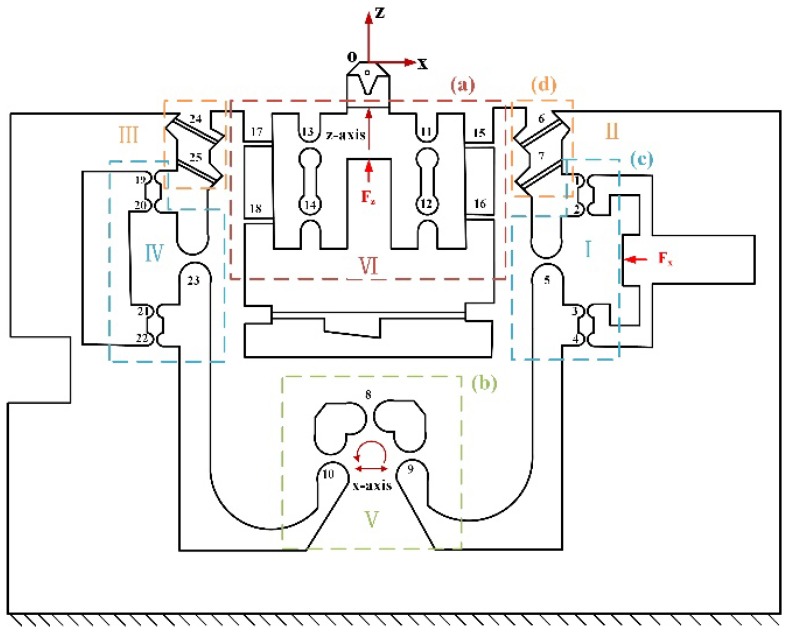
Schematic of the designed 2-DOF FTS.

**Figure 3 micromachines-10-00337-f003:**
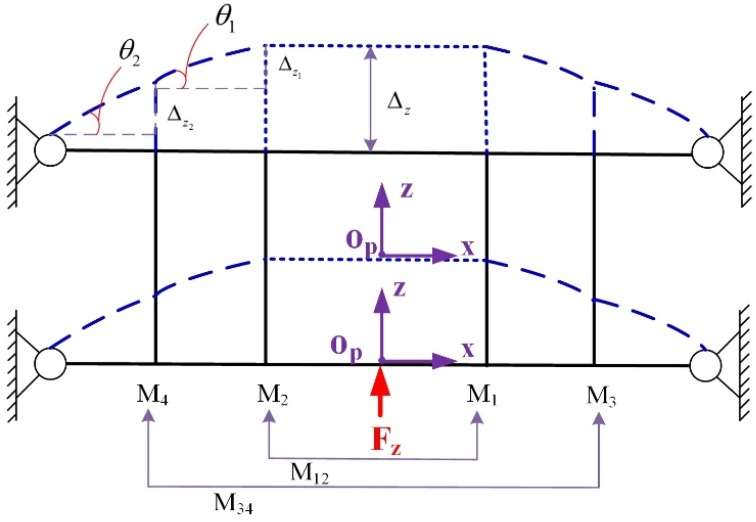
Schematic of moving principles of the high-rigidity four bar (HRFB) mechanism.

**Figure 4 micromachines-10-00337-f004:**
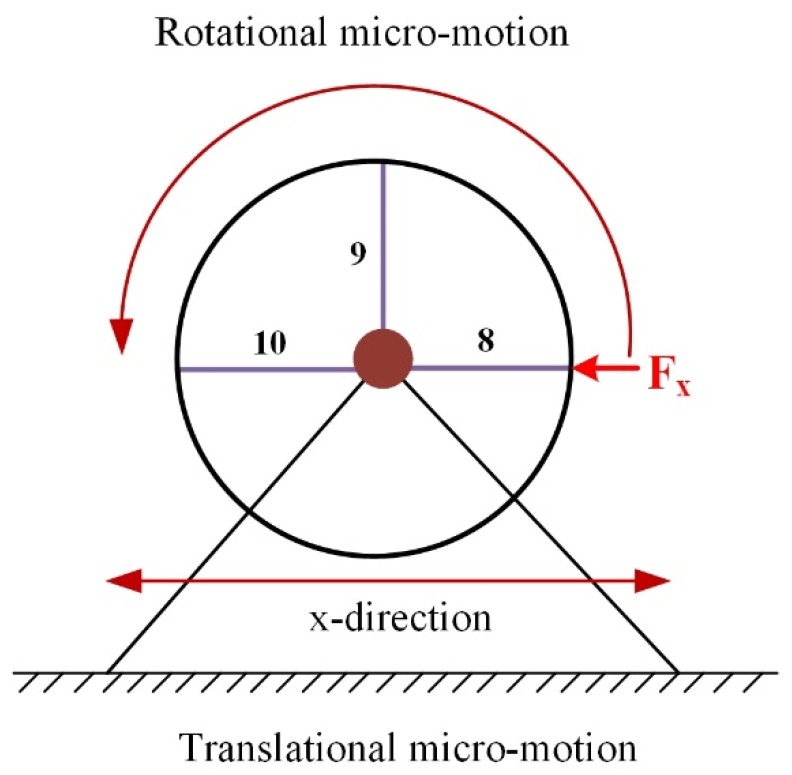
Schematic of moving principles of the flexible bearing hinges (FBHs) mechanism.

**Figure 5 micromachines-10-00337-f005:**
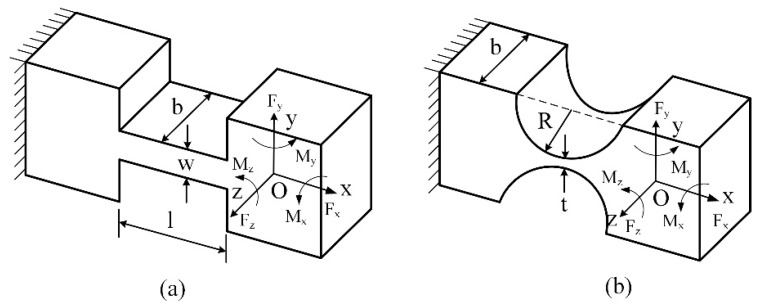
The coordinate system of (**a**) leaf spring flexible hinges and (**b**) right circular flexible hinges.

**Figure 6 micromachines-10-00337-f006:**
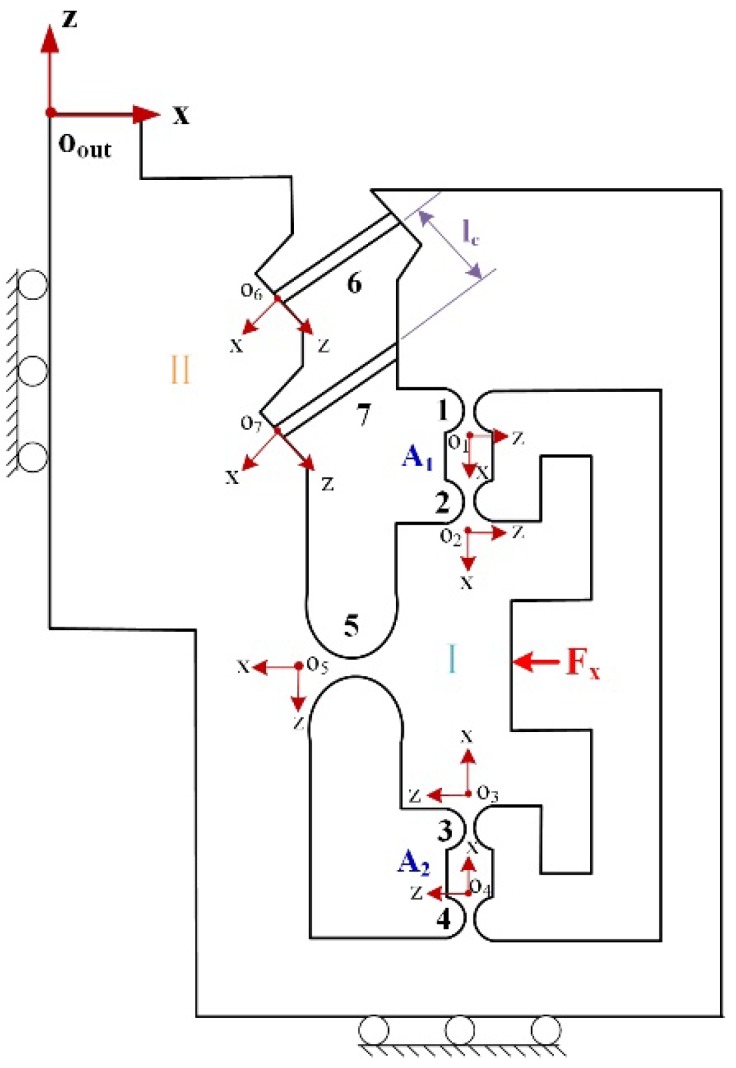
Schematic of module I and II.

**Figure 7 micromachines-10-00337-f007:**
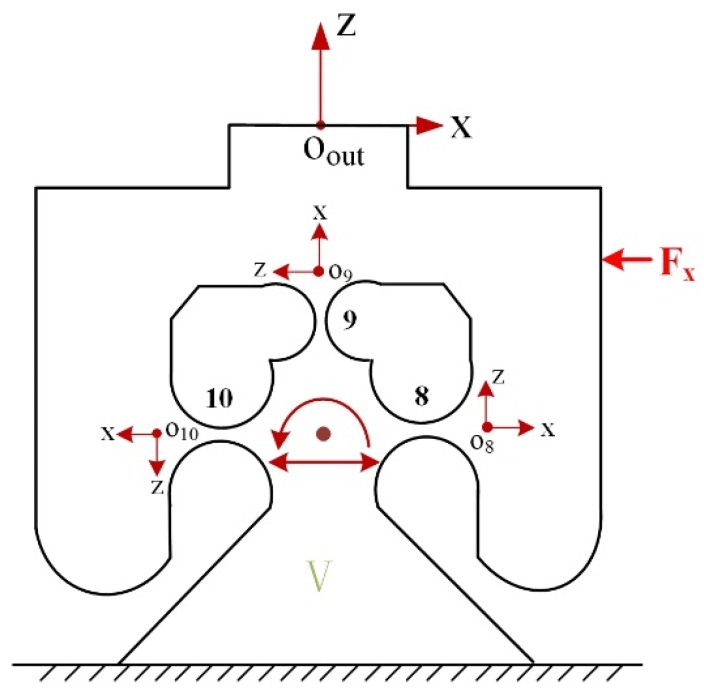
Schematic of the module V.

**Figure 8 micromachines-10-00337-f008:**
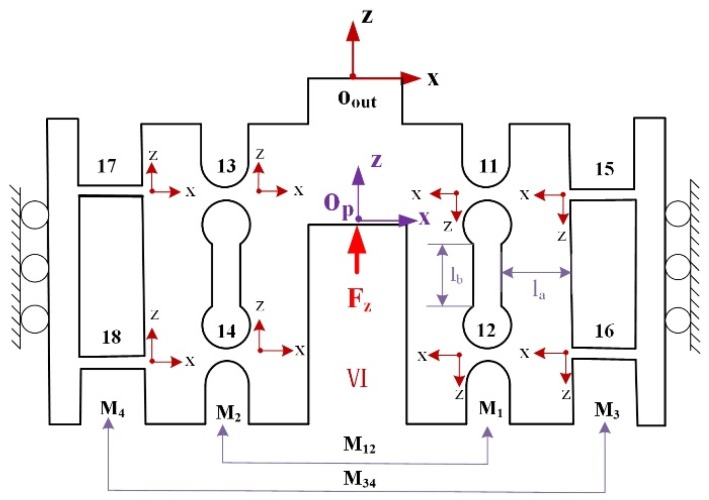
Schematic of the module VI.

**Figure 9 micromachines-10-00337-f009:**
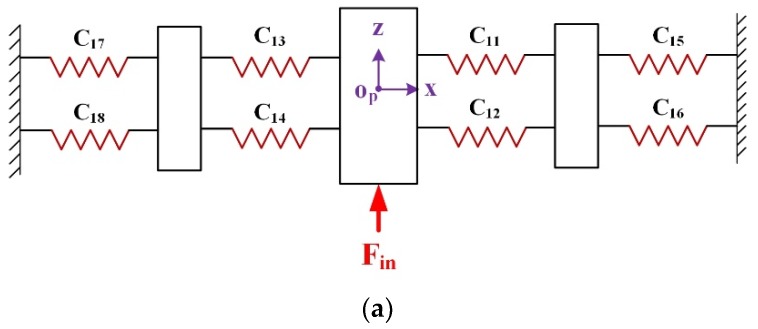
Input compliance model of the HRFB in the y-axis direction. (**a**) Equivalent spring model of the HRFB; (**b**) The simplified model of the HRFB.

**Figure 10 micromachines-10-00337-f010:**
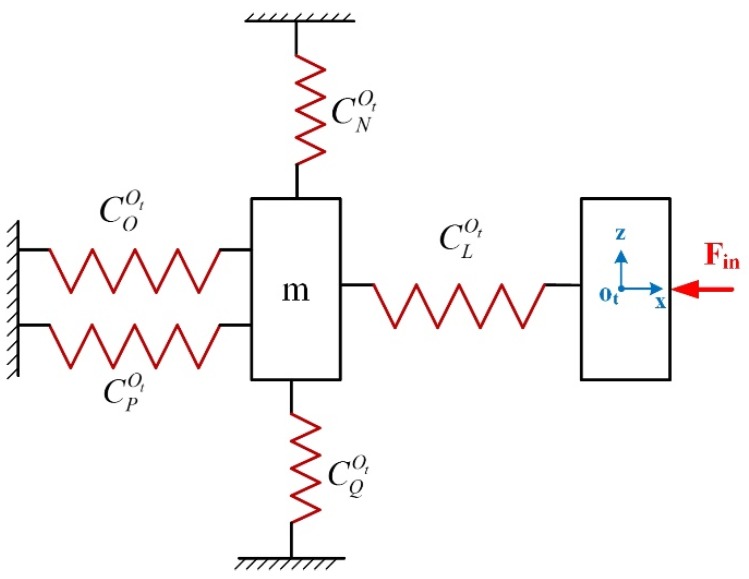
The simplified model of the input compliance in the x-axis direction.

**Figure 11 micromachines-10-00337-f011:**
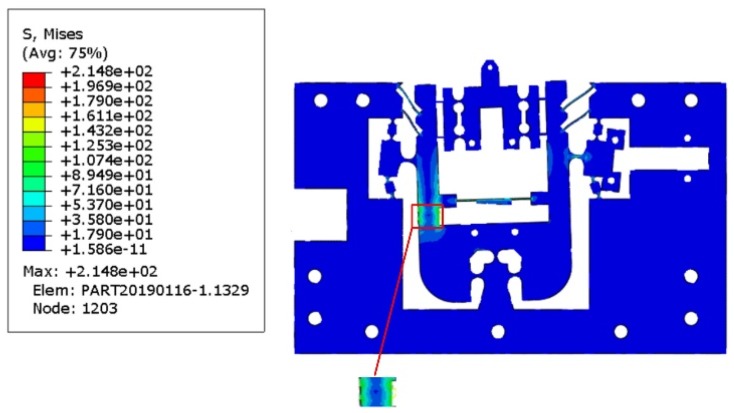
Stress simulation of the FTS.

**Figure 12 micromachines-10-00337-f012:**
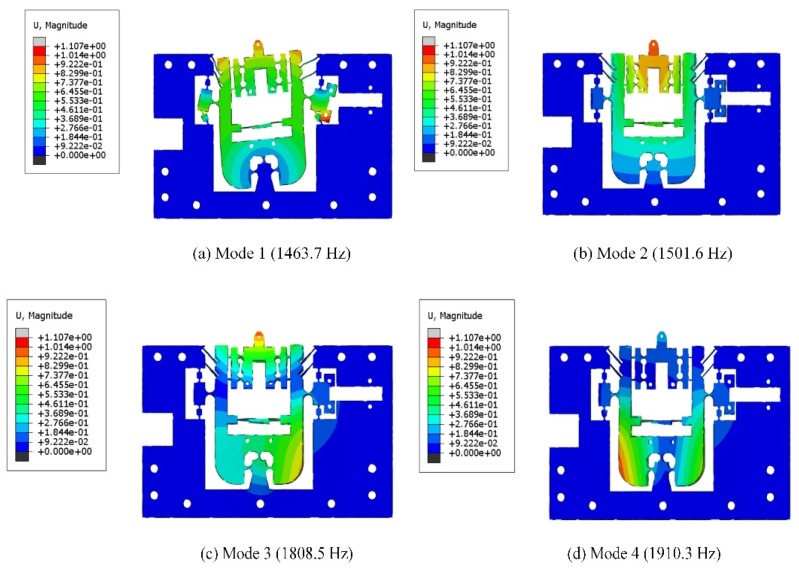
The first four modes of the mechanism.

**Figure 13 micromachines-10-00337-f013:**
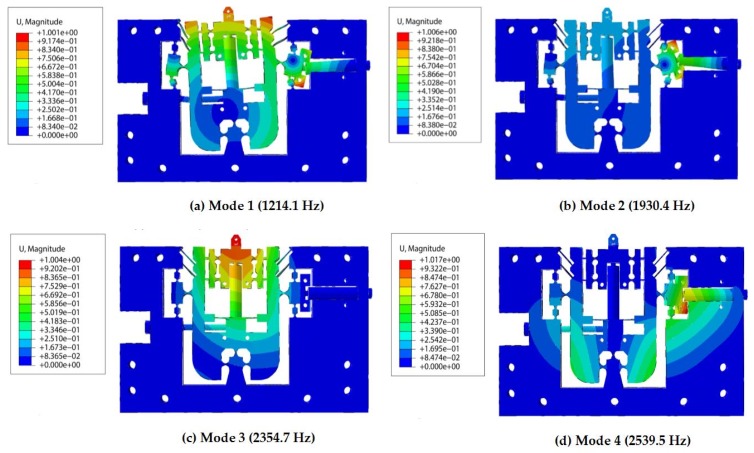
The first four modes of FTS assembly.

**Figure 14 micromachines-10-00337-f014:**
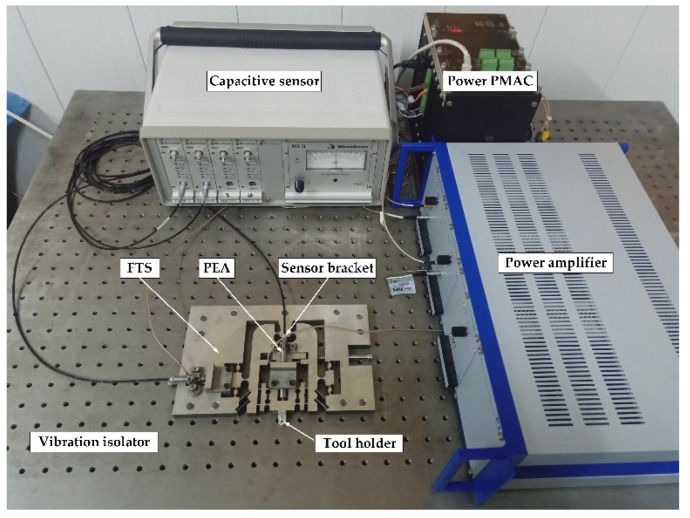
Experimental setup for performance testing.

**Figure 15 micromachines-10-00337-f015:**
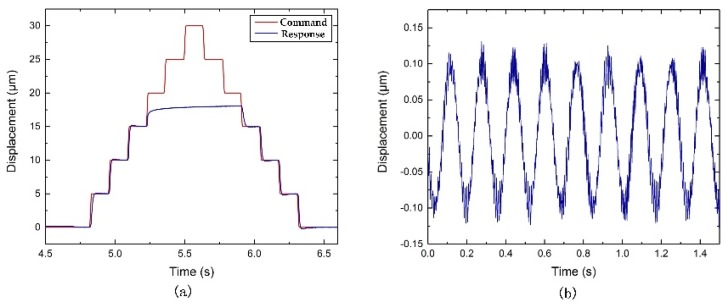
Experimental results of (**a**) motion stroke of the z-axis, (**b**) the coupling motion along the x-axis.

**Figure 16 micromachines-10-00337-f016:**
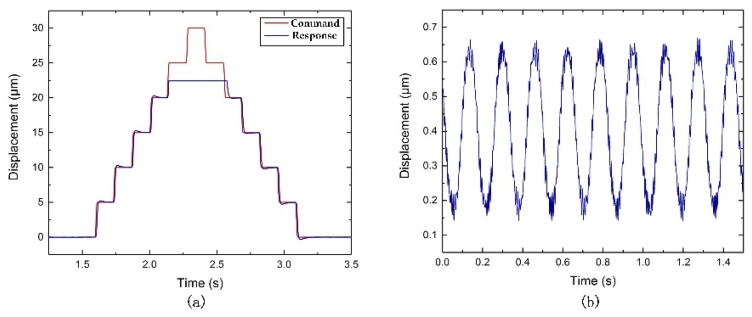
Experimental results of (**a**) motion stroke of the x-axis, (**b**) the coupling motion along the z-axis.

**Figure 17 micromachines-10-00337-f017:**
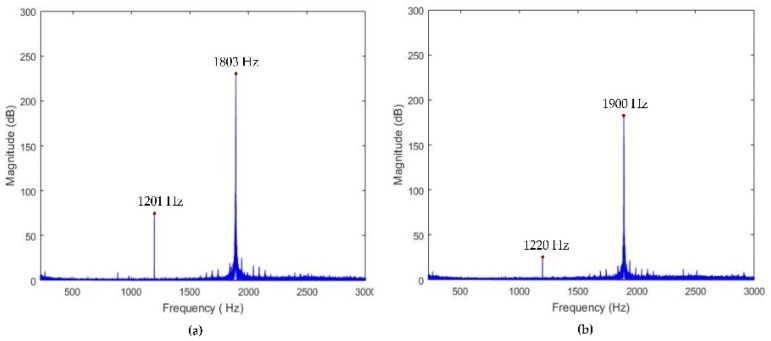
Dynamic responses along (**a**) z-axis and (**b**) x-axis.

**Figure 18 micromachines-10-00337-f018:**
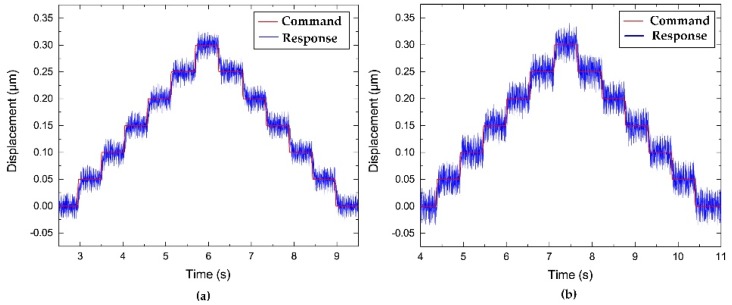
Resolutions of the mechanism along (**a**) z-axis and (**b**) x-axis.

**Figure 19 micromachines-10-00337-f019:**
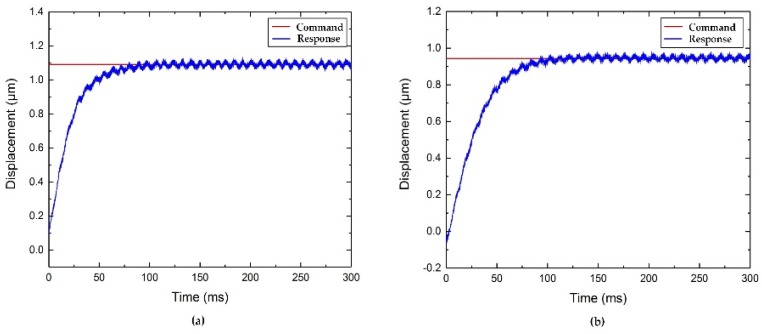
Step responses along (**a**) z-axis and (**b**) x-axis.

**Figure 20 micromachines-10-00337-f020:**
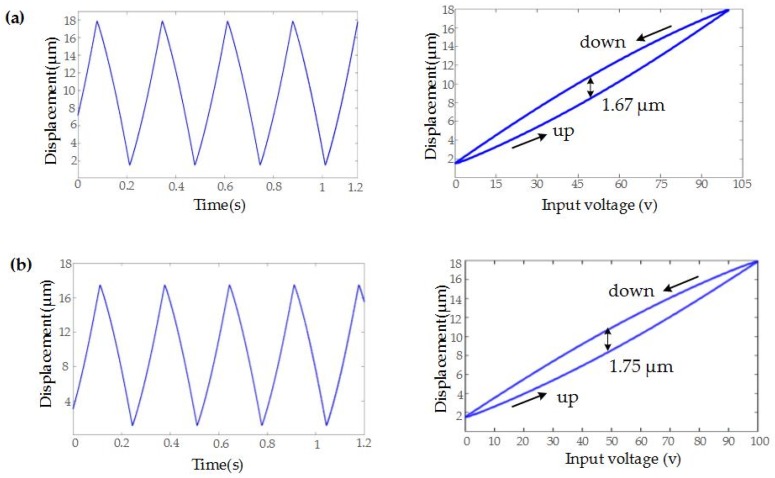
Hysteresis characteristic curve: (**a**) z-axis direction; (**b**) x-axis direction.

**Figure 21 micromachines-10-00337-f021:**
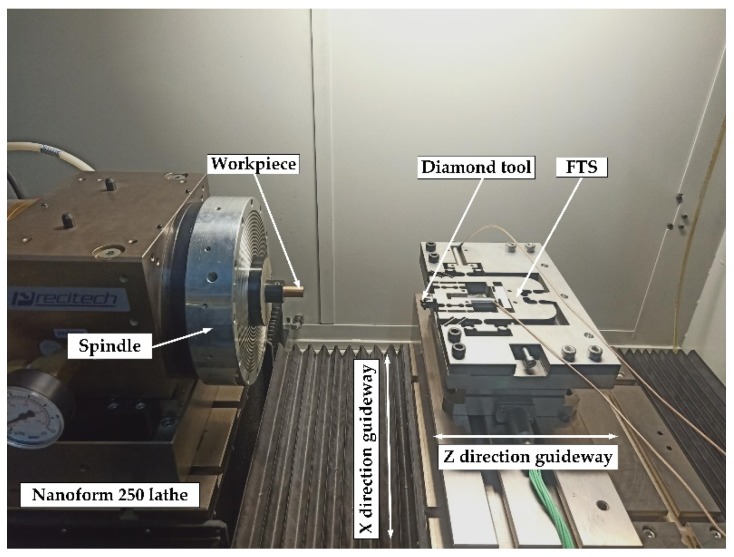
Experimental setup for the cutting.

**Figure 22 micromachines-10-00337-f022:**
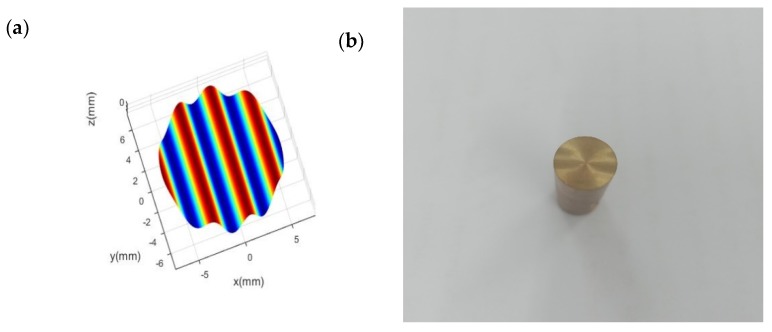
(**a**) Simulated sinusoidal wavy surface. (**b**) The sinusoidal wavy surface machined by the FTS mechanism.

**Figure 23 micromachines-10-00337-f023:**
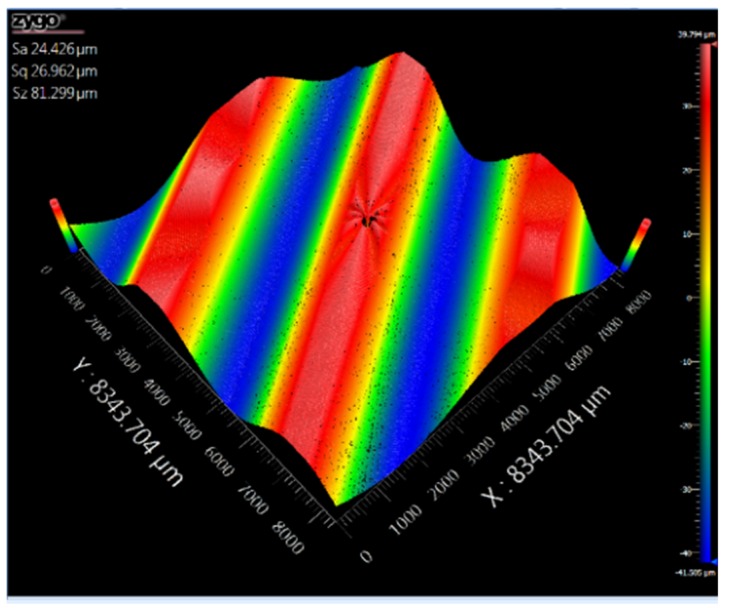
Sinusoidal wavy surface morphology.

**Table 1 micromachines-10-00337-t001:** Overall dimensional and material 7075 Al parameters of the mechanism.

*L*/mm	*B*/mm	*W*/mm	*E*/GPa	*σ*/MPa	*μ*	*ρ*/(kg·m^−3^)
154	246	12	71.7	503	0.33	2810

**Table 2 micromachines-10-00337-t002:** Main parameters of the high-rigidity four-bar (HRFB) mechanism.

*l_a_*/mm	*l_b_*/mm	*l*_1_/mm	*w*_1_/mm	*R*_1_/mm	*t*_1_/mm	*b*/mm
7.5	6.7	4	1	3.25	0.8	12

**Table 3 micromachines-10-00337-t003:** The dimensions of the other parts.

*l_c_*/mm	*l*_2_/mm	*w*_2_/mm	*R*_2_/mm	*t*_2_/mm	*R*_3_/mm	*t*_3_/mm	*R*_4_/mm	*t*_4_/mm
7.3	17.5	0.8	2	1	4	1.2	5	1.2

**Table 4 micromachines-10-00337-t004:** Performances evaluated by analysis model and finite element analysis (FEA) results.

Performance	Input Stiffness (x) (N/μm)	Input Stiffness (z) (N/μm)	Output Compliance (N/μm)
Matrix model	20.85	28.44	12.25
FEA	18.12	31.33	15.32
Deviation (%)	13.09	10.16	25.06

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
