# Peer review of "Development of Piezo-Actuated Two-Degree-of-Freedom Fast Tool Servo System"

_micromachines, 2019, doi:10.3390/mi10050337_

Round 1

Reviewer 1 Report

Paper title:

Development of piezo-actuated two-degree-of freedom fast tool servo system

The coauthors of this article designed and implemented a two degree of freedom fast tool servo (FTS), however, the reviewer's comments are as follows:

1-    Line 47, please explain to the reader: what are the non-resonant vibration cutting, and the resonant vibration cutting. Kindly mention the advantages and disadvantages of each type.

2-    Figure 1, the PZT used in the x-direction has no pre-tightening screw, please explain

3-    Line 227, in the static analysis, the material used in simulating the stresses was not defined

4-    Line 245, in the dynamic analysis, please define the expected exciting frequencies during the actual cutting

5-    Line 268, in the stroke and decoupling tests, the expected hysteresis of the strokes has not been studied, its effects on the motion trajectory have to be considered

6-    Line 302, step response test, please give more explanations and define the response mathematically

7-    Line 307, the experimental setup, please compare your results with previously published results and/or the results if the machine controller is only used   

Author Response

Dear reviewer,

We have made detailed revisions based on your comments. The point-by-point response can be found in the attachment.

Reviewer 2 Report

The paper developed a piezoelectric actuated device for the assistance of machining. A completed study was carried out, including analytical, FEA and experiments. The experiments have shown the effectiveness of the developed devices.

The reviewer would like to provide a few suggestions for the author’s consideration.

(1)   In line 55, give a few words to explain why y-axis is not unnecessary. In experiments, do you measure the displacement in y direction?

(2)    In Fig. 1, suggest give an overall dimension in three directions, as properties are affected by the dimension.

(3)    In static analysis, the stress analysis is done under loaded or no-load condition? Is it the same?

(4)    In section 3.4.2, there are four modes, it is first mode, second mode, or first-order mode, second-order mode?

(5)    In section 3.4.3, in simulation, actuator is not included, will it change the vibration mode?

(6)    Provide some information on piezo actuator.

(7)    Line 258, suggest using “4.1 Experimental setup”

(8)    In Fig. 12 (a) and 13 (a), the command and response line are not exactly the same, give some explanation.

(9)   In section 5.2, what exactly the signal applied to the actuator, x and z are in phase or not? Sa=24.426 um means what, the results is good, acceptable, improved?

Author Response

(The authors gave the same response as above.)

Reviewer 3 Report

To carry out two degree of freedom fast tool machining, this paper has developed experimental results to validate their designs. There remains several points to improve:

In the 6th line of Abstract, "product" should be written as "produce".

The 11th line of Introduction writes "Gao et al." in reference [14], which is not consistent with References in the manuscript.

Accompanied with Equations (3), (5), (9), etc. "Where" should be corrected as "where".

The frequency spectrum shown in Figure 14 should be compared with dynamic analysis results by using the finite element method.

Author Response

(The authors gave the same response as above.)

Round 2

Reviewer 1 Report

The required revisions have been performed.